# Cytomegalovirus-Specific T Cells in Pediatric Liver Transplant Recipients

**DOI:** 10.3390/v15112213

**Published:** 2023-11-04

**Authors:** Songpon Getsuwan, Nopporn Apiwattanakul, Chatmanee Lertudomphonwanit, Suradej Hongeng, Sophida Boonsathorn, Wiparat Manuyakorn, Pornthep Tanpowpong, Usanarat Anurathapan, Kanchana Tangnararatchakit, Suporn Treepongkaruna

**Affiliations:** 1Division of Gastroenterology, Department of Pediatrics, Faculty of Medicine, Ramathibodi Hospital, Mahidol University, Bangkok 10400, Thailand; songpon.get@mahidol.edu (S.G.); chatmanee.puk@gmail.com (C.L.); pornthep.tan@mahidol.ac.th (P.T.); 2Ramathibodi Excellence Center for Organ Transplantation, Faculty of Medicine, Ramathibodi Hospital, Mahidol University, Bangkok 10400, Thailand; 3Division of Infectious Disease, Department of Pediatrics, Faculty of Medicine, Ramathibodi Hospital, Mahidol University, Bangkok 10400, Thailand; sophida.bon@mahidol.ac.th; 4Division of Hematology and Oncology, Department of Pediatrics, Faculty of Medicine, Ramathibodi Hospital, Mahidol University, Bangkok 10400, Thailand; suradej.hon@mahidol.ac.th (S.H.); usanarat.anu@mahidol.ac.th (U.A.); 5Division of Allergy, Department of Pediatrics, Faculty of Medicine, Ramathibodi Hospital, Mahidol University, Bangkok 10400, Thailand; wiparai.man@mahidol.ac.th; 6Division of Nephrology, Department of Pediatrics, Faculty of Medicine, Ramathibodi Hospital, Mahidol University, Bangkok 10400, Thailand; kanchana.tan@mahidol.ac.th

**Keywords:** cytomegalovirus, liver transplantation, children, immunology, T cells

## Abstract

Cytomegalovirus (CMV) infection is a major opportunistic infection after liver transplantation (LT) that necessitates monitoring. Because of the lack of studies in children, we aimed to investigate CMV-specific T cell immune reconstitution among pediatric LT recipients. The recipients were monitored for CMV infection and CMV-specific T cells from the start of immunosuppressive therapy until 48 weeks after LT. Clinically significant CMV viremia (csCMV) requiring preemptive therapy was defined as a CMV load of >2000 IU/mL. Peripheral blood CMV-specific T cells were analyzed by flow cytometry based on IFNγ secretion upon stimulation with CMV antigens including immediate early protein 1 (IE1) Ag, phosphoprotein 65 (pp65) Ag, and whole CMV lysate (wCMV). Of the 41 patients who underwent LT, 20 (48.8%) had csCMV. Most (17/20 patients) were asymptomatic and characterized as experiencing CMV reactivation. The onset of csCMV occurred approximately 7 weeks after LT (interquartile range: 4–12.9); csCMV rarely recurred after preemptive therapy. Lower pp65-specific CD8+ T cell response was associated with the occurrence of csCMV (*p* = 0.01) and correlated with increased viral load at the time of csCMV diagnosis (ρ = −0.553, *p* = 0.02). Moreover, those with csCMV had lower percentages of IE1-specific CD4+ and wCMV-reactive CD4+ T cells at 12 weeks after LT (*p* = 0.03 and *p* = 0.01, respectively). Despite intense immunosuppressive therapy, CMV-specific T cell immune reconstitution occurred in pediatric patients post-LT, which could confer protection against CMV reactivation.

## 1. Introduction

Children who undergo liver transplantation (LT) generally receive immunosuppressive therapy targeting various immunologic mechanisms, particularly T cell function, to prevent graft rejection. However, this can contribute to an increased incidence of opportunistic infections caused by several viruses, including cytomegalovirus (CMV) [1]. CMV infection can lead to significant morbidity and mortality in LT recipients [2]. Clinical manifestations of CMV infection include enterocolitis, pneumonitis, retinitis and hepatitis [3]. Although the infection can be asymptomatic, it is also associated with impaired hepatic function and graft dysfunction [1]. Understanding the pathophysiology of CMV infection could aid treatment and prevent infection, thereby avoiding the associated negative outcomes.

Several researchers have studied the mechanisms of CMV infection after LT. Cell-mediated immunity, particularly CD4+ and CD8+ T cell responses, plays an important role in the prevention and resolution of CMV infection [2]. The development of CMV-specific T cells is a major mechanism of suppressing CMV infection after LT [4]. Unfortunately, most studies have been performed in adults. We aimed to investigate the roles of CMV-specific T cells and their potential clinical implications among pediatric LT recipients. We hypothesized that CMV-specific T cell immune reconstitution may occur in pediatric patients who undergo LT, even in those receiving immunosuppressive therapy.

## 2. Materials and Methods

### 2.1. Study Population and Monitoring of CMV Infection

Patients aged 0.5–18 years who underwent LT participated in the study at an active LT center in Bangkok, Thailand, from 31 October 2018 to 29 October 2019, 4 December 2020 to 3 December 2021, and 7 December 2022 to 7 August 2023. Peripheral blood specimens were collected from the enrolled subjects before and after LT (at weeks 0, 2, 4, 8, 12, 16, 24, 32 and 48), and CMV viral load was monitored by COBAS^®^ Ampliprep/COBAS TaqMan^®^ CMV Test *(Roche Diagnostics; Mannheim, Germany)*, which had a lower limit of detection of 56 IU/mL. Clinically significant CMV viremia (csCMV) requiring preemptive therapy was defined as a CMV load of >2000 IU/mL regardless of clinical symptoms [5]. Asymptomatic patients with csCMV were diagnosed as experiencing CMV reactivation, whereas patients with CMV diseases were those who had a CMV load above the detection limit along with any end-organ involvement, according to the American Society of Transplantation Infectious Diseases Community of Practice guidelines [3]. In this cohort, ganciclovir therapy (5 mg/kg/dose intravenously every 12 h) was implemented as a preemptive therapy for CMV reactivation and as a therapeutic treatment for CMV diseases.

The study was approved by the Human Research Ethics Committee at our institute, in accordance with the Helsinki Declaration (MURA2018/652).

### 2.2. Analysis of CMV-Specific T Cells

CMV-specific T cells were identified as in a previous study [6]. Venous blood was collected in a heparinized tube at room temperature and 3 mL of collected blood was subjected to separation of cellular portion from plasma portion by centrifugation at 3000× *g* for 10 min. The cellular portion was then used for mononuclear cell isolation which was performed by density gradient centrifugation using Lymphoprep^®^ (Axis-Shield; Oslo, Norway). The cellular portion was added with 4 mL of the reagent and centrifuged at 4 °C at 900× *g* for 30 min. The isolated mononuclear cells were washed with phosphate-buffered saline (PBS) (Sigma Aldrich, Burlington, MA, USA) and suspended in RPMI 1640 (GIBCO, Billings, MT, USA) medium containing 10% fetal bovine serum, FBS (GIBCO, Billings, MT, USA). The cells were then cultivated in tissue culture plates (Costar; Corning, NY, USA) at 2 × 105 cells in 200 uL per well. Then, CMV-specific viral antigens (Ags), including immediate early protein 1 (IE1) Ag, phosphoprotein 65 (pp65) Ag, (both from PepMix peptide pools, JPT Peptide Technologies; Berlin, Germany) and whole CMV lysate (wCMV) (Acris Antibodies; Rockville, MD, USA), were added. The final concentrations of IE1 and pp65 were 1 μg/mL and that of wCMV was 5 μg/mL. The cells in wells without Ag and those with phorbol myristate acetate (12.5 ng/mL)/ionomycin (2 μg/mL) were served as negative and positive controls, respectively. The cells were incubated at 37 °C in humidified air with 5% CO_2_ for 12 h. Brefeldin A (eBioscience; San Diego, CA, USA) at a final concentration of 10 μg/mL was then added and cells were further incubated at the same condition for 4 h. The cells were washed with PBS and resuspended in 1% formaldehyde (Sigma-Aldrich; St. Louis, MO, USA) for cell fixation. The cells were incubated for 15 min at room temperature. The cells were washed with PBS and then permeabilized in 0.5% saponin (Sigma-Aldrich; St. Louis, MO, USA) for 15 min at room temperature. The cells were washed with PBS and then incubated in the antibody cocktail (eBioscience; San Diego, CA, USA), consisting of FITC-CD3, PE-CD56, APC-CD4, eFluor780-CD8 and PE-Cy7 IFNγ at the final concentration of 1:250 at 4 °C devoid of light for 30 min. The cells were washed with PBS, resuspended in 100 μL of PBS and subjected to flow cytometry. The cell samples were acquired on a FACSVerse Flow cytometer (BD Biosciences). Data were analyzed by FlowJo version 10.8.0 (FlowJo LLC, Ashland, OR, USA). For gating strategy, lymphocyte population was gated first from FSC and SSC plot. CD3+ cells were then gated from the lymphocyte population, then CD4+ and CD8+ cells were gated from CD3+ cells.

CMV-specific T cells were defined as those secreting IFNγ upon stimulation with viral Ag. The percentage of CMV-specific T cells was reported as the proportion of IFNγ-producing cells among total CD4+ or CD8+ T cells.

### 2.3. Statistical Analyses

We used STATA (StataCorp, version 14, College Station, TX, USA) to perform statistical analyses. The Wilcoxon matched-pairs signed-rank test and Mann–Whitney test were applied to compare the variables of interest. The serial measurement of viral-specific T cells is displayed graphically. The linear mixed model with a random-effect covariance structure was used to study the serial measurements after natural logarithmic transformation. The Spearman rank correlation was calculated for correlation analyses. Coefficients (ρ) were classified as weak (ρ ≤ 0.3), moderate (0.3 < ρ ≤ 0.5), or strong (ρ > 0.5) correlation. Differences for which *p* < 0.05 were considered statistically significant.

## 3. Results

### 3.1. Patient Characteristics

A total of 41 patients participated in the study, with a median (interquartile range (IQR)) age of 1.9 (1.2–5.0) years. Most participants were diagnosed with biliary atresia (33 children; 80.5%) and underwent living-donor LT (34 children; 82.9%) (Table 1). All children were diagnosed with protein energy malnutrition before LT. The participants received ABO-compatible livers from their donors and received the same immunosuppressive regimen, consisting of steroids, tacrolimus, and mycophenolate mofetil (MMF) (Appendix A). Universal CMV prophylaxis was not implemented at our institute.

Most patients (38 patients; 92.7%) were positive for donor anti-CMV immunoglobulin G (IgG) and recipient anti-CMV IgG (D+/R+). One patient with D+/R- serostatus was diagnosed with primary CMV infection at 2 weeks after LT and received ganciclovir treatment for 4 weeks. A diagnosis of csCMV was reported in 20 children (48.8%), all of whom were D+/R+. Among these patients, 17 children were asymptomatic and, therefore, diagnosed with CMV reactivation; CMV enterocolitis was suspected in 3 patients who had diarrhea with positive stool CMV polymerase chain reaction tests. However, colonoscopy was not performed. The median (IQR) time of csCMV onset was 7 (4–12.9) weeks after LT, and duration of CMV treatment was 4 (3–4.1) weeks. Most patients had one episode of csCMV, and only two children had recurrent csCMV.

### 3.2. Peripheral Immune Cells in Enrolled Patients

Patients with csCMV tended to have lower percentages of peripheral CD4+ T cells during follow-up; the differences compared with the levels in patients without csCMV reached statistical significance at 2 weeks and 16 weeks after LT (median (IQR): 15.9% (11.7–19.3) versus 23.9% (16.1–31), *p* = 0.03, and 20.8% (14.2–27.7) versus 28.0% (23.1–35.0), *p* = 0.02, respectively). No significant differences were found in CD8+ T cell proportions. Among the patients with csCMV, seven (35%) were diagnosed with acute cellular rejection.

### 3.3. CMV-Specific T Cells Responding to pp65, IE1 and Whole Viral Lysate Ag in Patients Who Had Undergone LT

In the first 16 weeks after LT, children with positivity for anti-CMV IgG who subsequently developed csCMV tended to have a lower pp65-specific CD8+ T cell response than children without csCMV (*p* = 0.01) (Figure 1e) according to the linear mixed model. However, viral-specific CD4+ T cells did not differ between the csCMV and non-csCMV groups during the course of follow-up, except for lower proportions of IE1-specific CD4+ and wCMV-reactive CD4+ T cells in the csCMV group at 12 weeks after LT (0% (IQR: 0–0.002) versus 0.005% (IQR: 0–0.023) of total CD4+ T cells, *p* = 0.03, and 0% (IQR: 0–0.016) versus 0.021% (IQR: 0.008–0.058) of total CD4+ T cells, *p* = 0.01, respectively) (Figure 1a and Figure 1c, respectively). Among the patients in the csCMV group, we observed that the proportion of IE1-specific CD4+ T cells increased at approximately 36 weeks after infection (0.010% (IQR: 0–0.143) of total CD4+ T cells) (Figure 2a), whereas wCMV-reactive CD4+ and pp65-specific CD8+ T cell proportions increased during the earlier period, approximately 20 weeks (0.012% (IQR: 0–0.020) of total CD4+ T cells) (Figure 2b) and 24 weeks (0.032% (IQR: 0–0.045) of total CD8+ T cells) (Figure 2c) after infection, respectively.

### 3.4. Correlation between CMV Load at csCMV Diagnosis and Virus-Specific Peripheral Immune Cells before Diagnosis

In patients with csCMV, the percentage of pp65-specific CD8+ T cells before diagnosis of the infection negatively correlated with the viral load at diagnosis (ρ = −0.553, *p* = 0.02). No significant relationship was noted when comparing the viral load and total lymphocyte counts or other peripheral immune cells (Table 2).

### 3.5. CMV-Specific T Cells in Patients with Recurrent csCMV

Two patients developed recurrent csCMV in this study after receiving intense immunosuppressive therapy in addition to the general protocol. One patient had a history of late acute cellular rejection requiring high-dose corticosteroid therapy, high-dose tacrolimus, and MMF. Another patient, who underwent LT because of hepatoblastoma, had recurrent disease and subsequently received an additional course of chemotherapy. Unfortunately, the disease was uncontrolled; the patient and his caregivers elected to pursue palliative care and did not participate in further peripheral blood studies. Figure 3 illustrates the changes in CMV-specific T cells in the first patient. We noted that IE1-specific CD4+ T cells were initially present before the onset of csCMV and then disappeared after intense immunosuppression. Although these cells were not found after the first episode of csCMV, they were found after the second episode. We also observed that, although wCMV-reactive CD4+, wCMV-reactive CD8+, and IE1-specific CD8+ T cells responded well after the first episode of csCMV, the patient still developed recurrent csCMV after the second episode of intense immunosuppressive therapy.

## 4. Discussion

Approximately half of the children undergoing LT in this cohort had csCMV. Most were asymptomatic (CMV reactivation) and underwent preemptive antiviral therapy. This study demonstrated that CMV-specific T cell immune reconstitution occurs even in young post-LT children who received immunosuppressive therapy, although the magnitude of reconstitution was small. Moreover, patients with csCMV had a lower proportion of CMV-specific cytotoxic T cells than those without csCMV. Thus, we propose that T cells against several CMV Ags may be beneficial for the control and monitoring of CMV infection after pediatric LT.

After examining the roles of virus-specific CD8+ T cells, we found that children with csCMV tended to have fewer pp65-specific CD8+ T cells than those without csCMV. The peak response was observed around 24 weeks after LT. However, the onset of CMV viremia was around 7 weeks after LT. These findings may reflect a low-magnitude virus-specific T cell response that begins at the start of viremia and reaches its maximal response later during infection. We demonstrated that the levels of pp65-specific CD8+ T cells present before csCMV diagnosis strongly correlated with CMV loads. These findings highlight the importance of pp65-specific CD8+ T cells, which can be activated after transplantation [7]. Since CD8+ T cells are considered cytotoxic, a high number of these cells could suppress CMV replication. Nebbia et al. [8] also found that a level of pp65-specific CD8+ T cells >0.4% had high negative predictive value (92%) for CMV replication. A recent study in adults undergoing solid organ transplant [4] also noted that pp65-specific CD8+ T cells had an area under the receiver operator characteristic curve of 0.66 for the prevention of CMV infection. However, only a small proportion of patients (2.7%) in that study underwent LT.

Virus-specific CD4+ and CD8+ T cells need to function together to successfully combat CMV infection [9,10]. In this study, the levels of IE1-specific CD4+ and wCMV-reactive CD4+ T cells were significantly lower in patients with csCMV at 12 weeks after LT than in patients without csCMV. These results suggest that virus-specific CD4+ T cells respond to CMV infection, although we did not observe differences in CD4+ T cell levels at other time points. This could be due to the low magnitude of the CD4+ T cell responses in these patients, which made differentiation between the groups difficult. Despite an absence of statistically significant findings in a past study [11], positive IE1-specific CD4+ T cells have been posited to play a role in adults undergoing LT. Shin et al. [12] demonstrated that the presence of IE1-specific CD4+ T cells before LT was a negative predictor of CMV viremia within 2 months after LT. In our patient with recurrent csCMV, we noted that IE1-specific CD4+ T cells levels did not increase until the second episode of csCMV, after which he did not develop another reinfection. However, the rise of other virus-specific T cells after the first episode of csCMV seemed to be inadequate to prevent recurrent csCMV. In addition, Nebbia et al. [8] demonstrated that the presence of wCMV-reactive, IL-2–producing CD4+ T cells before LT, especially at a level of >0.1% of total CD4+ T cells, was a negative predictor of CMV viremia. Another study in adults suggested roles for other CMV-specific CD4+ cells, including pp65-specific CD4+ cells, as negative predictors of CMV infection after LT [13]. In this study, the failure to demonstrate that the three types of virus-specific CD4+ T cells could predict the likelihood of CMV viremia after LT may be attributable to our patients having quite poor immune responses from a young age and experiencing chronic malnutrition, rendering them unable to mount appropriate cellular immune responses.

Nevertheless, our study highlighted that pediatric LT recipients, even young patients on immunosuppressive drugs, produce some CMV-specific CD4+ T cells. The lack of a statistical difference in CD4+ T cell levels in patients with and without csCMV may be due to the small number of these cells that was detected, which limited our power to differentiate between the two groups.

There were some limitations to our study. The small number of participants could affect the robustness of the statistical analyses. As this study was performed in young children, the results of positive anti-CMV IgG may be caused by the detection of maternal antibodies in children aged less than 1 year. Moreover, the study was performed during the SARS-CoV-2 (COVID-19) pandemic, which caused difficulties in terms of specimen collection. Blood samples could not be obtained from some participants at certain time points because of travel restrictions. Although this study suggested that CMV-specific T cells, such as pp65-specific CD8+, IE1-specific CD4+, and wCMV-reactive CD4+ T cells, may have potential clinical implications, caution is warranted when using these cells to monitor very young children. Burton et al. [14] suggested that virus-specific T cells were less sensitive for CMV risk stratification among children aged less than 12 years. Nevertheless, several reports have proposed that virus-specific T cells could be a better option for risk stratification or the prediction of future CMV infection, which could lead to more appropriate management strategies [15,16,17].

In conclusion, this study successfully demonstrated that CMV-specific T cell immune reconstitution occurs in young pediatric LT recipients. Further investigation is needed to explore the roles these cells play in the pathophysiology of CMV infection in very young individuals.

## Figures and Tables

**Figure 1 viruses-15-02213-f001:**
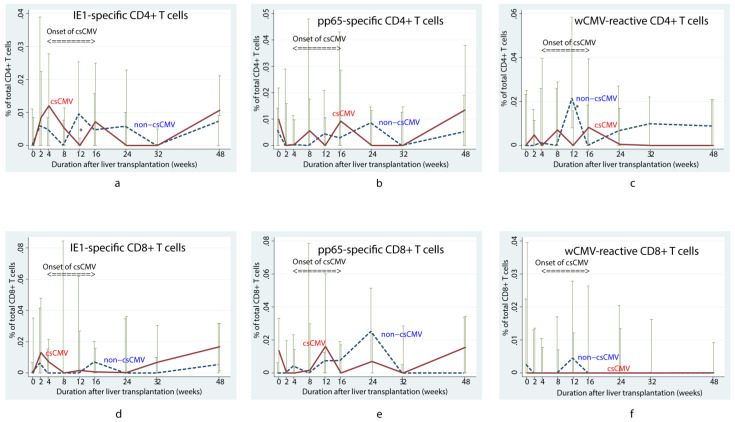
Changes in median percentage of cytomegalovirus-specific T cells over time in children who underwent liver transplantation and were positive for anti-cytomegalovirus immunoglobulin G According to the mixed linear model, the children with clinically significant CMV viremia (csCMV) tended to have fewer pp65-specific CD8+ T cells than those who did not develop viremia (*p* = 0.01) in the first 16 weeks (**e**). However, changes of other specific T cells were not significantly different between both groups, including IE1-specific CD4+ (**a**), pp65-specific CD4+ (**b**), wCMV-reactive CD4+ (**c**), IE1-specific CD8+ (**d**), and wCMV-reactive CD8+ T cells (**f**). Lower proportions of IE1-specific CD4+ and wCMV-reactive CD4+ T cells in the csCMV group at 12 weeks after LT were noted as asterisks in (**a**,**c**), respectively. csCMV: clinically significant CMV viremia; IE1: early protein 1; pp65: phosphoprotein 65; wCMV: whole CMV lysate.

**Figure 2 viruses-15-02213-f002:**
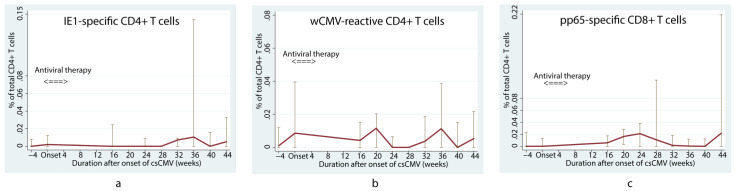
Changes in median percentage of cytomegalovirus-specific T cells in children who underwent liver transplantation with positive anti-cytomegalovirus immunoglobulin G and developed clinically significant cytomegalovirus viremia. Changes in the median percentage of cytomegalovirus (CMV)-specific T cells include IE1-specific CD4+ (**a**), wCMV-reactive CD4+ (**b**), and pp65-specific CD8+ T cells (**c**), over time among children who had anti-CMV immunoglobulin G, and later developed clinically significant CMV viremia. The proportion of IE1-specific CD4+ T cells increased at approximately 36 weeks after infection (**a**), whereas wCMV-reactive CD4+ and pp65-specific CD8+ T cell proportions increased during the earlier period, approximately 20 weeks (**b**) and 24 weeks (**c**) after infection. csCMV: clinically significant CMV viremia; IE1: early protein 1; pp65: phosphoprotein 65; wCMV: whole CMV lysate.

**Figure 3 viruses-15-02213-f003:**
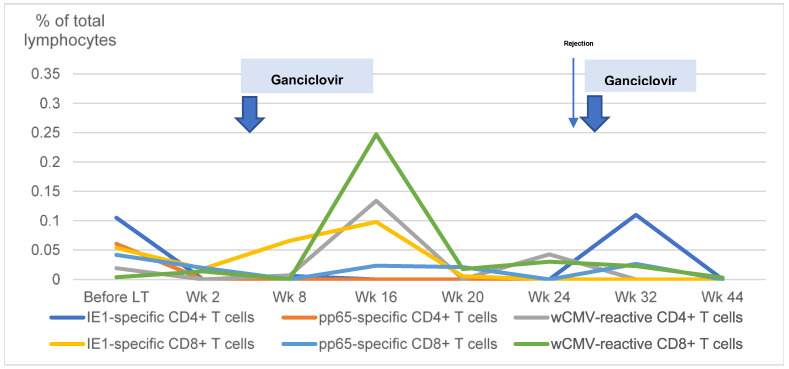
Changes in cytomegalovirus-specific T cells in a patient with recurrent clinically significant cytomegalovirus viremia. The changes in cytomegalovirus (CMV)-specific T cells in a child with recurrent clinically significant CMV viremia is demonstrated. LT: liver transplantation; Wk: weeks; IE1: early protein 1; pp65: phosphoprotein 65; wCMV: whole CMV lysate.

**Table 1 viruses-15-02213-t001:** Characteristics of participants in the study (N = 41).

Characteristics	Results
Age (years), median (IQR)	1.9 (1.2–5.0)
Male, N (%)	21 (51.2%)
Diagnosis of primary liver disease, N (%)	
- Biliary atresia	33 (80.5%)
- Other chronic liver diseases *	6 (14.6%)
- Acute liver failure	2 (4.9%)
Types of liver transplantation, N (%)	
- Living donor liver transplantation	34 (82.9%)
- Deceased donor liver transplantation	7 (17.1%)
Acute cellular rejection, N (%)	11 (26.8%)
Anti-CMV IgG before transplant, N (%)	
- D+/R+	38 (92.7%)
- D+/R−	1 (2.4%)
- D−/R+	2 (4.9%)
Clinically significant CMV viremia after transplant, N (%)	20 (48.8%)
- CMV reactivation	17
Onset of CMV infection after transplant (weeks), median (IQR)	7 (4–12.9)
Duration of CMV infection treatment (weeks), median (IQR)	4 (3–4.1)

* Alagille syndrome (N = 2), autoimmune sclerosing cholangitis (N = 1), non-syndromic bile duct paucity (N = 1), Budd–Chiari syndrome (N = 1), Hepatoblastoma (N = 1). CMV: cytomegalovirus; IQR: interquartile range; IgG: immunoglobulin G; D: donor; R: recipient.

**Table 2 viruses-15-02213-t002:** Correlation between cytomegalovirus load at the diagnosis of clinically significant cytomegalovirus viremia and the percentage of cytomegalovirus-specific T cells per total lymphocytes before the diagnosis (N = 20).

Cells	Spearman’s Rho	*p*
**Total lymphocyte count**	0.406	0.10
**Percentage of cells per total CD4+ T cells**		
- IE1-specific CD4+	−0.097	0.70
- pp65-specific CD4+	0.039	0.88
- wCMV-reactive CD4+	0.125	0.62
**Percentage of cells per total CD8+ T cells**		
- IE1-specific CD8+	−0.298	0.23
- pp65-specific CD8+	−0.553	0.02
- wCMV-reactive CD8+	−0.273	0.27

CMV: cytomegalovirus; IE1: early protein 1; pp65: phosphoprotein 65; wCMV: whole CMV lysate.

## Data Availability

The data presented in this study are available on request from the corresponding author. The data are not publicly available unless the Faculty of Medicine Ramathibodi Hospital permits.

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
