# Peer review of "Cytomegalovirus-Specific T Cells in Pediatric Liver Transplant Recipients"

_viruses, 2023, doi:10.3390/v15112213_

Round 1
Reviewer 1 Report
Comments and Suggestions for Authors
Getsuwan et al investigated the kinetics of CMV-specific T-cell responses in paediatric liver transplant recipients and compared responses between those who developed clinically significant CMV to those who did not. The paper is well written and findings from the study will contribute to the knowledge gap in this research field.
Minor comments:
1. Lines 84-87 – include clone details for antibodies and the cytometer used for the experiment.
2. Lines 99-100 – this should be included in the ‘Study population and monitoring of CMV infection’ section.
3. Lines 175-178 – there are additional rows of Spearman’s correlation and p-values that do not line up with the cell populations of interest. Please ensure that the interpretation of the results is still correct once this have been rectified.
4. Line 232 – ‘T cells’ instead of ‘cells’
5. Figures 1 and 2 – Fonts for x- and y-axis labels are too small.
6. Figure 3 – The words ‘ganciclovir’ and ‘rejection’ in the text boxes have been cut off.
Author Response
Reviewer Comments, Author Responses and Manuscript Changes
Reviewer 1
Getsuwan et al investigated the kinetics of CMV-specific T-cell responses in paediatric liver transplant recipients and compared responses between those who developed clinically significant CMV to those who did not. The paper is well written and findings from the study will contribute to the knowledge gap in this research field.
Comments
- Lines 84-87 – include clone details for antibodies and the cytometer used for the experiment.
- Response: We added the clone details of antibodies and cytometer used for the experiment in page 3 line 96-105
- The cells were washed with PBS and then incubated in the antibody cocktail (eBioscience; San Diego, CA, USA), consisting of FITC-CD3 (clone HIT3a), PE-CD56 (clone CMSSB), APC-CD4 (clone OKT4), eFluor780-CD8 (clone RPA-T8) and PE-Cy7 IFNγ (clone 4S.B3) at the final concentration of 1:250 at 4oC devoid of light for 30 minutes. The cells were washed with PBS, resuspended in 100 uL of PBS and subjected to flowcytometry. The cell samples were acquired on a FACSVerse Flow cytometer (BD Biosciences). Data were analyzed by FlowJo version 10.8.0 (FlowJo LLC, USA). For gating strategy, lymphocyte population was gated first from FSC and SSC plot. CD3+ cells were then gated from the lymphocyte population, then CD4+ and CD8+ cells were gated from CD3+ cells.
- Changes: We have included more details on the analysis of CMV-specific T cells on page 2, line 73. The clone details for antibodies and the cytometer were briefly mentioned on page 3, line 96.
- Lines 99-100 – this should be included in the ‘Study population and monitoring of CMV infection’ section.
- Response: We agree with the suggestion.
- Changes: The details of ethical consideration have been moved to the ‘Study population and monitoring of CMV infection’ section on page 2, line 71.
- Lines 175-178 – there are additional rows of Spearman’s correlation and p-values that do not line up with the cell populations of interest. Please ensure that the interpretation of the results is still correct once this have been rectified.
- Response: Our apologies for the technical errors that the Spearman’s correlation and p-values did not line up with the cell populations of interest.
- Changes: The table has been modified on page 6, line 198.
- Line 232 – ‘T cells’ instead of ‘cells’
- Response: We agree with the correction.
- Changes: ‘T cells” was added on page 7, line 25
- Figures 1 and 2 – Fonts for x- and y-axis labels are too small.
- Response: Thank you for the suggestion
- Changes: We have increased the font size in the x- and y-axis of both figures.
- Figure 3 – The words ‘ganciclovir’ and ‘rejection’ in the text boxes have been cut off.
- Response: We also notice the errors during the production period.
- Changes: Both words in figure 3 have been modified.

Reviewer 2 Report
Comments and Suggestions for Authors
In the manuscript “Cytomegalovirus-specific T cells in pedriatic liver transplant recipients” the authors were able to detect CMV-specific T cell responses in children after liver transplantation under immunosuppression. Only for pp65-specific CD8+ T cells they were able to detect a significant relation between percentage of positve cells and clinically relevant CMV reactivation. Due to a low number of patients (however, high for this special patient group!) the results are difficult to interprete. Nevertheless, the results of this well done study are worthwhile to be published.
I have some minor questions/suggestions:
1. Most constellations were D+/R+. Did you consider the possibility of detection of maternal antibodies in the youngest children (0.5 to 1 year)?
2. What was the course of the patient with the constellation D+/R-? I would expect primary CMV infection. Please comment shortly in the results part.
3. Please adapt the scale in figure 2 to figure 1. It is very difficult to interprete the data with the different scales.
4. In figure 1 there are 3 asterisks (1a and 1c). I think those above week eleven should mark lower proportions of cells in one group. However, the information is lacking in the figure legend. I did not find any information to the asterisk in 1c, week 48.
5. Discussion, 7, lines 236 and 237: CMV reactivation, reinfection is wrong in this case.
Author Response
Reviewer 2
In the manuscript “Cytomegalovirus-specific T cells in pedriatic liver transplant recipients” the authors were able to detect CMV-specific T cell responses in children after liver transplantation under immunosuppression. Only for pp65-specific CD8+ T cells they were able to detect a significant relation between percentage of positive cells and clinically relevant CMV reactivation. Due to a low number of patients (however, high for this special patient group!) the results are difficult to interpret. Nevertheless, the results of this well-done study are worthwhile to be published.
Comments
- Most constellations were D+/R+. Did you consider the possibility of detection of maternal antibodies in the youngest children (0.5 to 1 year)?
- Response: We agree that children aged less than 1 year could have positive anti-CMV IgG due to their maternal antibodies. Therefore, this could be one of the limitations of this study.
- Changes: The limitation has been added on page 8, line 273.
- What was the course of the patient with the constellation D+/R-? I would expect primary CMV infection. Please comment shortly in the results part.
- Response: One patient with D+/R- serostatus was diagnosed with primary CMV infection at 2 weeks after LT and received ganciclovir treatment for 4 weeks.
- Changes: The results of one patient with D+/R- serostatus were described on page 4, line 132.
- Please adapt the scale in figure 2 to figure 1. It is very difficult to interprete the data with the different scales.
- Response: Thank you for the suggestion.
- Changes: The scale in figure 2 has been modified.
- In figure 1 there are 3 asterisks (1a and 1c). I think those above week eleven should mark lower proportions of cells in one group. However, the information is lacking in the figure legend. I did not find any information to the asterisk in 1c, week 48.
- Response: Lower proportions of IE1-specific CD4+ and wCMV-reactive CD4+ T cells in the clinically significant CMV viremia (csCMV) group at 12 weeks after LT were noted as asterisks in figure 1a and 1c, respectively. The third asterisk was mistakenly presented in the previous version of the manuscript.
- Changes: The details of asterisks in figure 1 were added to the figure legend on page 5, line 173. The third asterisk has been removed from figure 1c.
- Discussion, 7, lines 236 and 237: CMV reactivation, reinfection is wrong in this case.
- Response: We initially would like to discuss that the rise of other virus-specific T cells after the first episode of csCMV seemed to be inadequate to prevent another episode of csCMV.
- Changes: “The first infection” was replaced by “the first episode of csCMV”, and “reinfection” was changed to “recurrent csCMV” in page 8, line 257.
